# Transstadial Transmission from Nymph to Adult of *Coxiella burnetii* by Naturally Infected *Hyalomma lusitanicum*

**DOI:** 10.3390/pathogens9110884

**Published:** 2020-10-25

**Authors:** Julia González, Marta G. González, Félix Valcárcel, María Sánchez, Raquel Martín-Hernández, José M. Tercero, A. Sonia Olmeda

**Affiliations:** 1Villamagna S.A., Finca ‘‘La Garganta’’, 14440 Villanueva de Córdoba, Spain; julia.gonzalez@rutgers.edu (J.G.); marta.gonzalez@inia.es (M.G.G.); maria2985@hotmail.com (M.S.); jmtercero@fincalagarganta.com (J.M.T.); 2Center for Vector Biology, Department of Entomology, Rutgers University, New Brunswick, NJ 08901, USA; 3Grupo de Parasitología Animal, Animalario del Departamento de Reproducción Animal, INIA, 28040 Madrid, Spain; 4Bee Pathology laboratory, Centro Apícola Regional, JCCM, 19180 Marchamalo, Spain; rmhernandez@jccm.es; 5Departamento de Sanidad Animal, Facultad de Veterinaria, UCM, 28040 Madrid, Spain; angeles@ucm.es

**Keywords:** Q fever, tick, meso-Mediterranean, transstadial transmission, artificial feeding

## Abstract

*Coxiella burnetii* (Derrick) Philip, the causative agent of Q fever, is mainly transmitted by aerosols, but ticks can also be a source of infection. Transstadial and transovarial transmission of *C. burnetii* by *Hyalomma lusitanicum* (Koch) has been suggested. There is a close relationship between this tick species, wild animals and *C. burnetii* but the transmission in a natural environment has not been demonstrated. In this study, we collected 80 engorged nymphs of *H. lusitanicum* from red deer and wild rabbits. They moult to adults under laboratory conditions and we feed them artificially through silicone membranes after a preconditioning period. *C. burnetii* DNA was tested in ticks, blood and faeces samples using real-time PCR. The pathogen was found in 36.2% of fed adults, demonstrating that transstadial transmission from nymph to adult occurs in nature. The presence of DNA in the 60.0% of blood samples after artificial feeding confirms that adults transmit the bacteria during feeding. Further studies are needed about co-feeding and other possible transmission routes to define the role of this tick species in the cycle of *C. burnetii*.

## 1. Introduction

*Coxiella burnetii* (Derrick 1939) Philip 1948 is the agent of Q fever, an important zoonotic disease with a broad range of hosts involved [1]. Different modes of transmission have been reported, ticks being a source of infection for animals [1,2,3,4]. However, its role in the *C. burnetii* cycle could depend on the tick species, among other factors. For example, in Northern Spain, where *Hyalomma* spp ticks were not found, *C. burnetii* DNA was detected in a low rate in ticks [5,6], while a high number of ticks were positive in Central Spain [7,8]. In this latest area, wild animals as red deer (*Cervus elaphus* L.) and wild rabbits (*Oryctolagus cuniculus* L.) seem to be more important as a source of infection than domestic animals, and *Hyalomma lusitanicum* (Koch) ticks are suspected to be the vector of this pathogen in the habitat [7,8,9,10]. Transstadial and transovarial transmission of *C. burnetii* by this tick species has been suggested, but its vector capacity has yet to be confirmed.

Artificial tick feeding systems provide a very useful tool to study pathogen transmission [11,12] because they allow monitoring of the feeding process under laboratory conditions, avoiding the use of experimental animals [13]. In this study, we tested the transstadial transmission nymph to adult of *H. lusitanicum* and discuss possible routes of transmission by naturally infected adults, analyzing the blood and faeces samples obtained during artificial feeding.

## 2. Results and Discussion

Regarding the detection limit of the technique, dilutions up to 10 genome equivalents (GE) of *C. burnetti* showed an identifiable threshold cycle (C_T_), indicating that our method allowed us to detect low *C. burnetti* loads. Although a range of IS1111 copies can be present in the Coxiella-like endosymbionts of ticks [14], the sequencing of the selected positive amplicons confirmed the specific detection of *C. burnetti*. Negative extraction controls excluded cross contamination of the DNA processing step.

We detected *C. burnetii* DNA in 36.2% of newly moulted adult ticks. This prevalence was higher than the prevalence levels reported for *H. lusitanicum* [7] and other tick species (Duron et al., 2015), usually below 10%. These results also coincide with the findings of a previous study where not all the ticks that fed on a *C. burnetii*-positive animal became infected [8], because not all ticks are susceptible to carrying the pathogen [4]. The detection of *C. burnetii* in newly moulted adults confirm the ability of naturally infected *H. lusitanicum* nymphs to maintain the bacteria during the moulting process.

In the artificial feeding assays, we recovered eight gravid females, six engorged females and 33 unengorged females (Table 1). We detected *C. burnetii* DNA in both sexes (46.8% of females and 21.2% of males) and at different degrees of engorgement (Table 2), because not all ticks feed at the same rate. Half of the gravid females obtained laid eggs after the blood meal and three out of four egg masses hatched, but no pool of larvae was *C. burnetii*-positive. These gravid females were processed as carcasses after the oviposition, which could have reduced the DNA burden of the pathogen in the detection.

Male ticks tend to feed little in the wild, and therefore, it is expected that they play a minor role in the maintenance and/or transmission of *C. burnetti*. However, several experimental studies have reported the role of male ticks as a possible reservoir and their ability to sexually transfer other pathogens to females [15,16,17,18], and the results of our study may suggest the importance of *H. lusitanicum* males (Table 2). It seems that a complete engorgement is not necessary to transmit the bacteria during feeding, because *C. burnetti* DNA detection was significantly higher (Chi-square (χ^2^) = 3.872, df = 1, *p* < 0.05) in ticks of feeding units where we did not recover gravid females (47.4% versus 26.2%) (Table 1).

The pathogen was detected in the blood samples of 60.0% of the feeding units, between six and eight days, and between 10 and 17 days (Figure 1). Unfortunately, we could not check if the bacteria were alive or dead. Nevertheless, the percentage of detection of *C. burnetti* DNA was significantly lower in those females that completed engorgement (Table 1), suggesting that the pathogen could interfere in the physiology of ticks. Thus, if the bacteraemia is low, the tick could transmit *C. burnetti* and survive, but when the bacteraemia is high, the tick could die due to the disturbance of its development (incomplete feeding), although it can transmit the pathogen. It is described that most *H. lusitanicum* adults attach to the membrane at days 5–6 and detach at day 11 on average during artificial feeding [19]. Thus, we detected *C. burnetti* DNA during the two feeding phases described in the tick pattern [20,21]: slow phase (attachment and engorgement) and rapid phase (full engorgement during the last 24 h). This suggests the transmission of *C. burnetti* by this tick species during the first days after attachment. The speed of infection is known for other agents but was not described for *C. burnetii* [22]. However, we did not detect the pathogen in faeces samples. It is reported that *C. burnetii* is found in the tick gut [4] and tick faeces [23]. A low bacteria count could explain the difficulty in detecting this pathogen in tick faeces, as it has been recently suggested [12]. We did not know the level of bacteraemia in the wild animals sampled, but the prevalence of *C. burnetii* in both animals and their ticks was high in a previous study [8]. The detection of *C. burnetii* DNA in those engorged nymphs collected from red deer and wild rabbit was 16.7% and 67.0%, respectively, and 38.0% of red deer’s livers and 43.4% of rabbit’s anal/vaginal swabs were positive. Körner et al. [12] reported the secretion via faeces as a possible route of transmission by ticks, and they did not always detect *C. burnetii* DNA on blood during artificial feeding experiments, indicating the need of high bacteraemia in the host for excretion. It is important to note that two daily blood exchanges (*Coxiella*-free blood in this study) are necessary to develop the artificial feeding system and the faeces obtained were maintained inside the feeding units until the end of the assays. These conditions could have underestimated the infection rate, whilst in nature ticks constantly feed on infected blood if the host is indeed infected.

In conclusion, the detection of *C. burnetii* in newly molted adults demonstrate that transstadial transmission from nymph to adult occurs in nature, and the presence of DNA in the blood samples after artificial feeding confirms that adults transmit the bacteria during feeding. Co-feeding transmission could have occurred in this study because ticks feed very close to one another inside the unit. This phenomenon has been described for several tick-borne viruses in addition to bacteria such as *Borrelia burgdorferi,* as infected and uninfected ticks feed in close proximity to one another on the same host [24]. However, this was difficult to demonstrate in this study since a priori we did not know which adults were negative or positive and the artificial feeding protocol requires exchanging blood twice daily, which could limit reinfection during feeding.

## 3. Materials and Methods 

### 3.1. Ticks

A total of 80 naturally engorged *H. lusitanicum* nymphs from red deer and wild rabbit were selected for this study, those with better conditions to moulting. The rest of the ticks were analyzed previously to evaluate the rate of *C. burnetii* infection among the wildlife and this tick species [8]. Red deer were examined after being hunted and wild rabbits were captured live and released at the capture site after the tick collection, under the permission of the Community Authority and maintaining suitable animal welfare. Animals were carefully examined to remove ticks using tweezers. Ticks were kept under laboratory conditions (25 °C and 85% relative humidity and a natural light–dark cycle), until moulting to adults during a preconditioning period of 2–8 months.

### 3.2. Artificial Tick Feeding

The experimental design was based on in vitro feeding with silicone membranes as described by Kröber and Guerin (2007) and adapted to *H. lusitanicum* by González et al. (2017). After the preconditioning period, adults were separated in groups of 6–10 (more than 50% females) per feeding unit. Sterile commercial defibrinated ovine blood was used (Oxoid, Madrid, Spain) with two exchanges daily. At the end of artificial feeding, ticks were recovered and classified by sex and degree of engorgement (unengorged females: light feeding, no signs of faeces in the feeding unit; engorged females: moderate feeding, making it possible to collect faeces; and gravid females: fully engorged). The gravid females were maintained under laboratory conditions until oviposition or/and hatching. All specimens, besides an aliquot of blood (1 mL) per every exchange, and the faeces from each feeding unit were stored at −20 °C prior to DNA extraction.

### 3.3. Sample Preparation and PCR Analysis

Ticks and faeces samples (considered as tissue) were processed according to the described methodology [8]. Similarly, each blood sample (250 μL) was mixed with 25 μL of protease (Qiagen, Hilden, Germany) and 250 μL of AL buffer (Qiagen), prior to being kept at 70 °C for 10 min. Then, 250 μL of isopropanol (Qiagen) and 35 μL of MagAttract Suspension G (Qiagen) were added to the mixture. Once samples were prepared, DNA was extracted using a BS96 DNA Blood Kit (Qiagen) following the protocols for tissue (ticks and faeces) or blood, respectively, in a BioSprint 96 workstation (Qiagen). Cross contamination during DNA extraction was excluded by running negative controls (nuclease-free water; GE Healthcare Life Sciences, Logan, UT, USA; one well for every 20 samples and per plate). Also, sterile commercial defibrinated ovine blood was analyzed to discard the presence of *C. burnetti* in the blood meal. All DNA samples were analyzed by real-time PCR targeting the transposase gene of the *C. burnetii*-specific IS1111a insertion element as previously described [25]. To confirm the *C. burnetti*-specific amplification, two positive samples (representing the 22% of positives detected) were randomly selected and sequenced by Sanger technology (at the Genomic Service of Complutense University, Madrid, Spain). Additionally, to determine our detection level, a standard curve was performed using a *C. burnetii*-positive control (strain Nine Mile phase II), kindly supplied by Dr. Jado (Carlos III Health Institute, Madrid, Spain). To do this, serial dilutions were made from 10^6^ genome equivalent (GE) per μL up to 10 GE/μL, and they were analyzed using two replicates per dilution. Samples were considered positive when a threshold cycle (C_T_) value was ≤40.

### 3.4. Data Analyses

Statistical analyses were performed by IBM SPSS Statistics 20 software [26]. Differences in the *C. burnetii* detection between the feeding units with feeding rate were estimated by Chi-square tests. Significance was set at *p*-value < 0.05.

## 4. Conclusions

These results demonstrate that *H. lusitanicum* ticks can transmit *C. burnetii* at least from the nymph to adult stage and newly molted adults are able to spread the bacteria to blood during feeding. These results also suggest the important role played by *H. lusitanicum* males in maintaining this pathogen. However, further research is needed to know if there are alternative transmission routes of *C. burnetii* by ticks (co-feeding and tick faeces) with experimental infections under biosecurity conditions.

## Figures and Tables

**Figure 1 pathogens-09-00884-f001:**
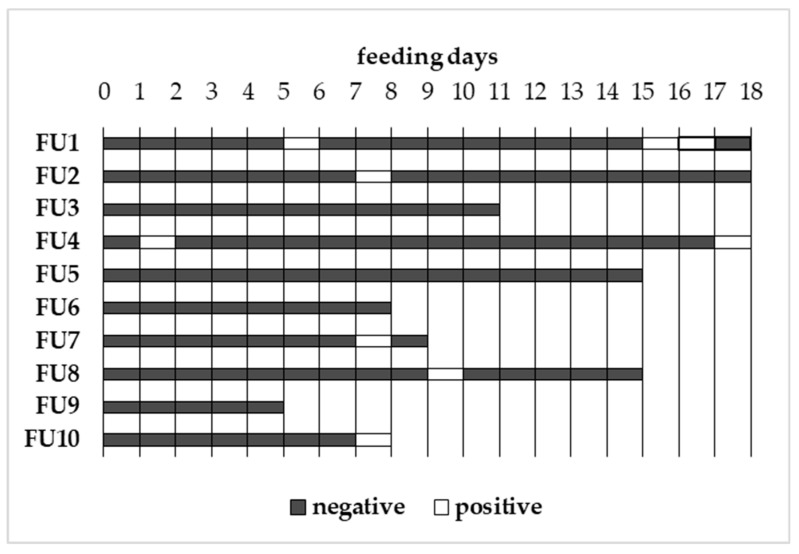
Duration of artificial feeding assays. *Coxiella burnetii*-positive blood samples are marked in white color in each feeding unit (FU) timeline and negative samples in grey color.

**Table 1 pathogens-09-00884-t001:** Feeding units used in the assays and ticks recovered after the artificial feeding.

Feeding Unit	Number of Ticks	Number of Females	Number of Males	Feeding Rate ^1^ (%)	Degree of Engorgement ^2^		
					GF	EF	UF
**1**	6	3	3	100.0	3		
**2**	10	6	4	33.3	2	1	3
**3**	8	4	4	25.0	1	1	2
**4**	8	4	4	25.0	1		3
**5**	10	6	4	16.7	1	1	4
**6**	8	4	4	0		1	3
**7**	7	5	2	0		1	4
**8**	9	6	3	0			6
**9**	7	4	3	0			4
**10**	7	5	2	0		1	4

^1^ Feeding rate: (number of gravid females/number of females) × 100. ^2^ Degree of engorgement on female: GF = gravid female; EF = engorged female; UF = unengorged female.

**Table 2 pathogens-09-00884-t002:** *Coxiella burnetii* DNA detection in the fed ticks artificially.

Feeding Units	Mean Feeding Rate ^1^ (%)	Positive Ticks (%)	Positive Females (%)	Positive Males (%)
**With GF (N = 5)**	40.0	26.2a	39.1a	10.5a
**Without GF (N = 5)**	0	47.4b	54.2a	35.7a
**Total**	**20.0**	**36.2**	**46.8**	**21.2**

^1^ Mean feeding rate: mean percentage of gravid females per females placed in the feeding units. GF = gravid females. Percentages with different letters are significantly different (χ^2^ test, *p* < 0.05).

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
