# Peer review of "Transstadial Transmission from Nymph to Adult of Coxiella burnetii by Naturally Infected Hyalomma lusitanicum"

_pathogens, 2020, doi:10.3390/pathogens9110884_

Round 1

Reviewer 1 Report

This is a very interesting study. 

The paper could be improved however.

  1. The use of IS1111a as the gene target for the detection of Coxiella burnetii is of concern, as this gene is present in some other Coxiella spp which are endosymbionts of ticks. It is not a foolproof marker of C.burnetii. This should be mentioned in the Discussion.
  2. The Abstract needs to make clear that the "blood samples" are from an artificial feeding apparatus, and not from an animal.
  3. The use of percentages to 2 decimal points is wrong. The data is not accurate to two decimal points ! In most cases a percentage calculation to the nearest whole number would be more appropriate. This is true in both the text and Table 1.
  4. Table 1 is confusing. It needs to be made clearer. Maybe make 2 separate Tables out of the "with feeding" and "without feeding" data.
  5. In Table 1 footnote, put the "GF=" explanation first and before using the abbreviation.
  6. In the Acknowledgements, I think the role of "His Grace, the Duke of Westminster" should be clarified. Was it financial support ? scientific support ?
  7. While the standard of English is good, it could be improved and I suggest a native English speaker go through the paper.

Author Response

We thank you the comments and suggestions provided by the reviewer, and we have followed all of them trying to make more clear the manuscript. 

We sincerely think that the manuscript has improved a lot.

We are going to provide a document explaining all the points indicated by the reviewer.

Reviewer 2 Report

Relationship of Coxiella burnetii and ticks have been gray zone for a long time. Questionnaire of this study is important to understand the disease Q fever. However, this manuscript needs to organize their experiments and results. Some critical information was not clearly described. How the authors confirmed “transstadial transmission” by testing only adult ticks? Detecting C. burnetii gene after feeding C. burnetii contain blood cannot explain “transstadial transmission” because DNA fragment can be detected after digestive concentration even the bacteria is transmitted deadly. 

Author Response

(The authors gave the same response as above.)

Reviewer 3 Report

Dear Author, this is a very clear manuscript, straightforward. And very interesting results, I have only very minor comments that I added directly in the pdf.

Author Response

We especially thank you the comments and suggestions provided by the reviewer, and we have followed all of them trying to make more clear the manuscript. 

We sincerely think that the manuscript has improved a lot.

We are going to provide a document explaining all the points indicated by the reviewer.

Round 2

Reviewer 2 Report

I am grateful to review revised version of the manuscript. Now their experiments and results are friendly to readers. This study provides better understanding of C. burnetii maintenance in nature.